# Does coronavirus disease 2019 history alone increase the risk of postoperative pulmonary complications after surgery? Prospective observational study using serology assessment

**Ah Ran Oh[1‡], Eun-Suk Kang[2‡], Jungchan Park[1], Sangmin Maria Lee[1], Mijeong Jeong[3], Jong-Hwan Lee[1] ***

1 Department of Anesthesiology and Pain Medicine, Samsung Medical Center, Sungkyunkwan University School of Medicine, Seoul, Korea, 2 Department of Laboratory Medicine and Genetics, Samsung Medical Center, Sungkyunkwan University School of Medicine, Seoul, Republic of Korea, 3 Research Institute for Future Medicine, R&D Management & Supporting Team, Republic of Korea

‡ ARO and ESK are contributed equally to this work as co-first authors.
* jonghwanlee75@gmail.com

**Data Availability Statement:** Data cannot be shared publicly because of security for data warehouse of our institution. Data are available

## Abstract

### Background

Concern exists about the increasing risk of postoperative pulmonary complications in patients with a history of coronavirus disease 2019 (COVID-19).

### Objective

We conducted a prospective observational study that compared the incidence of postoperative pulmonary complications in patients with and without a history of COVID-19.

### Methods

From August 2022 to November 2022, 244 adult patients undergoing major non-cardiac surgery were enrolled and allocated either to history or no history of COVID-19 groups. For patients without a history of confirming COVID-19 diagnosis, we tested immunoglobulin G to nucleocapsid antigen of SARS-CoV-2 for serology assessment to identify undetected infection. We compared the incidence of postoperative pulmonary complications, defined as a composite of atelectasis, pleural effusion, pulmonary edema, pneumonia, aspiration pneumonitis, and the need for additional oxygen therapy according to a COVID-19 history.

### Results

After excluding 44 patients without a COVID-19 history who were detected as seropositive, 200 patients were finally enrolled in this study, 100 in each group. All subjects with a COVID-19 history experienced no or mild symptoms during infection. The risk of postoperative pulmonary complications was not significantly different between the groups according

from the Samsung Medical Center Institutional Data Access / Ethics Committee (contact via Institutional Review Board of Samsung Medical Center, +82-02-3410-1808) for researchers who meet the criteria for access to confidential data.

**Funding:** The author(s) received no specific funding for this work.

**Competing interests:** The authors have declared that no competing interests exist.

to the history of COVID-19 (24.0% vs. 26.0%; odds ratio, 0.99; 95% confidence interval, 0.71–1.37; P-value, 0.92). The incidence of postoperative pulmonary complications was also similar (27.3%) in excluded patients owing to being seropositive.

## Conclusion

Our study showed patients with a history of no or mild symptomatic COVID-19 did not show an increased risk of PPCs compared to those without a COVID-19 history. Additional precautions may not be needed to prevent PPCs in those patients.

## Introduction

Since first reported in late 2019, the coronavirus disease 2019 (COVID-19), caused by the novel severe acute respiratory syndrome coronavirus 2 (SARS-CoV-2), has spread to become a global pandemic [1]. As of December 2022, the World Health Organization estimates that over 651 million people have been infected with COVID-19, with over 6 million cases of mortality [2]. Recently, however, the pandemic has been well-controlled in some regions through the implementation of public health measures, and in some areas, it is being changed into an endemic [3]. While public health authorities should continue monitoring the virus's spread and implement effective control measures to prevent future outbreaks, those changes mean we also need to pay attention to medical care in patients who have recovered from COVID-19.

Postoperative pulmonary complications (PPCs) are significant causes of morbidity and mortality after major non-cardiac surgery [4]. There has been concern that a COVID-19 history increases the risk of PPCs because the virus can cause various pulmonary symptoms, including severe respiratory illness [5–7]. In addition, several previous studies, mainly in active and just-resolved COVID-19 patients, showed potential increases in postoperative complications, including PPCs [5, 8–11]. However, few studies have evaluated the increases in PPCs according to the actual extent of COVID-19 history. Considering SARS-CoV-2 infection can frequently be undetected [12], results from previous analysis need to be more accurate.

Therefore, to exclude infected patients who had not been diagnosed, we employed a prospective design with the serological assessment to evaluate whether a COVID-19 history can increase the risk of PPCs. This approach can allow a more accurate comparison by overcoming the gap between seroprevalence and cumulative incidence, a limitation of previous analysis. We also expected the present analysis to provide directions for optimal perioperative care in patients with a COVID-19 history.

## Methods

The protocol for this prospective study was carefully reviewed and approved by the Institutional Review Board of Samsung Medical Center (SMC 2022-04-168-001) and was registered at https://cris.nih.go.kr (Registration number KCT0007530). The written informed consent was obtained from all study patients, who were fully informed of the nature and purpose of the study and any potential risks or benefits. The study was conducted in accordance with the Declaration of Helsinki.

## Study patients and serology assessment

From August 2022 and November 2022, we included adult patients undergoing an elective major non-cardiac non-thoracic surgery under endotracheal intubation. The patients enrolled for this study were evaluated for history of COIVD-19 infection, symptoms during infection, and number of vaccinations. In addition, all patients showed negative result in reverse transcriptase polymerase chain reaction (RT-PCR) test performed within 3 days before surgery. We defined the patients with history of positive result RT-PCR as confirmed cases of COVID-19 [13]. We also identified patients without history of COVID-19 as having negative result in immunoglobulin G (IgG) test for SARS-CoV-2 nucleocapsid antigen (anti-N antibody) [14, 15]. The detection of SARS-CoV-2 IgG antibody is an indicator of recovery and immunity from COVID-19, which implies that the patient was previously infected with SARS-CoV-2 [16]. For all patients reporting no history of confirmed diagnosis of COVID-19, we tested anti-N antibody using 3ml sample of blood during intraoperative period to identify undetected infection in the past. Then, only seronegative patients were included and classified as cases without a history of COVID-19. Electrochemiluminescence immunoassay kit the Elecsys® Anti-SARS-CoV-2 assay (Roche Diagnostics, Basel, Switzerland) was used to detect anti-N antibody qualitatively. Assays were performed according to the manufacturer's detailed instructions at Cobas e801 autoanalyzer (Roche Diagnostics, Basel, Switzerland). The result of a sample is reported with a cut-off index (COI), and the results of at least 1.0 COI were considered positive. The overall specificity of test is reported as 99.80% [17]. We continued this test until the number of seronegative patients met the estimated target sample size. Exclusion criteria included patients with intubation before surgery, active COVID-19 infection, and without preoperative chest x-ray. Finally, the incidences of PPCs were compared according to the confirmed history of COVID-19.

**Clinical management and definitions.**    Surgical and anesthetic managements followed the institutional protocol. During anesthesia, we used a standardized ventilator setting consisting of a tidal volume of 6–8 mL per ideal body weight with a positive end-expiratory pressure of 5 cm $H_2O$. A respiratory rate was adjusted to maintain normocapnia. The inspiratory:expiratory ratio was set at 1:2, and the $FIO_2$ was adjusted to maintain an oxygen saturation of greater than 95%. The ventilator was set to assist-control mode, with the patient's tidal volume and respiratory rate monitored continuously using a capnograph and pulse oximeter. Any necessary adjustments to the ventilator settings were made by the attending anesthesiologist.

Major elective operations included mastectomy, colorectal resection for cancer (colectomy and proctectomy), prostatectomy, pancreatic resections (Whipple, total pancreatectomy, and distal pancreatectomies), hepatectomy, gastrectomy, hip replacement, knee replacement, laminectomy, hysterectomy, spinal fusion, elective open repair of abdominal aortic aneurysm, elective endovascular repair of abdominal aortic aneurysm, brain tumor resection, and carotid endarterectomy [8]. PPCs were identified based on postoperative diagnoses or imaging findings within 30 days after surgery and included atelectasis, pleural effusion, pulmonary edema, pneumonia, aspiration pneumonitis, and need for additional oxygen therapy [4, 18].

## Statistical analysis

The baseline characteristics were expressed as a number (%) for categorical variables and as mean (± standard deviation) for continuous variables. Differences were presented using chi-square test or Fisher's exact test for categorical variables and Student's t-test or Mann–Whitney U-test for continuous variables, as appropriate. The risk of PPCs was compared with the logistic regression analysis and reported as odds (OR) with 95% confidence interval (CI). The variables for multivariable adjustment were selected with P-value < 0.15 or clinical relevance

which were age, male, the American Society of Anesthesiologists physical status, chronic kidney disease, and operation duration. All statistical analyses in this study were performed by R 4.2.0 (Vienna, Austria; http://www.R-project.org/).

## Sample size

The sample size of the study population was calculated using MedCalc 12.3.0.0 (MedCalc Software, Mariakerke, Belgium). According to our previous study [19], the incidence of PPCs after major non-cardiac surgery was reported to be 30%, and we assumed that the incidence of PPCs would increase up to 50% in patients with history of COVID-19. With 5% type I error rate and 80% power, 93 patients in each group were required. Considering a drop-out rate of 5%, we planned to include 200 patients for this study (100 patients in each group).

## Results

A total of 244 patients were initially enrolled for this study. Among the patients who reported no confirmed history of COVID-19, 44 patients were excluded from the primary analysis owing to seropositivity. Then, 200 patients were finally enrolled for primary analysis (Fig 1). Baseline characteristics according to COVID-19 history are summarized in Table 1. Patients with a history of COVID-19 tended to be younger and more male, but it was not statistically significant. The difference for chronic kidney disease was marginally significant with lower incidence in the group with a history of COVID-19. Baseline characteristics and incidence of PPCs of the entire population including seropositive patients are summarized in S1 Table.

The overall incidence of PPCs was 25% (50/200). The incidence of PPCs as well as each composite according to COVID-19 history is shown in Table 2 and Fig 2. After an adjustment, the risk of PPCs was not different between the groups (24% vs. 26%; OR: 0.99; 95% CI: 0.71–1.37; P-value = 0.92) (Table 2). In patients with a history of COVID-19, the incidences of PPCs according to disease-relevant factors such as duration between diagnosis and surgery, vaccination, and presence of symptoms were summarized in Table 3.

## Discussion

Our study comparing the risk of PPCs according to the presence of a COVID-19 history showed a history of no or mild symptomatic SARS-CoV-2 infection alone did not increase the incidence of PPCs. In addition, we confirmed SARS-CoV-2 infection history by the absence of antibodies assessed from a serology test in patients without a history of COVID-19. Those results seem timely as the COVID-19 pandemic continues to evolve, and the virus may be transitioning to an endemic state in some areas of the world.

Perioperative care has been a topic of particular importance during the COVID-19 pandemic [20–23]. In the early stages of the pandemic, studies focused on the clinical challenges of providing perioperative care during a pandemic, including maintaining a safe environment for patients and healthcare workers [20, 22]. More recent studies focused on the risk of postoperative complications in patients with COVID-19 [5, 8, 21, 24–27]. These studies have raised attention to patients with perioperative SARS-CoV-2 infection who are at higher risk of PPCs. However, recently, as the COVID-19 pandemic shifts towards an endemic state and becomes more widespread, a growing number of patients undergo surgery after completely recovering from SARS-CoV-2 infection. Therefore, it seems imperative to focus on these patients and set guidelines for their management during the perioperative period.

Many studies have investigated the postoperative outcomes in patients with COVID-19 in various settings, but some had fundamental limitations. Those studies analyzed the incidence of postoperative complications merely using patients with a COVID-19 history without a

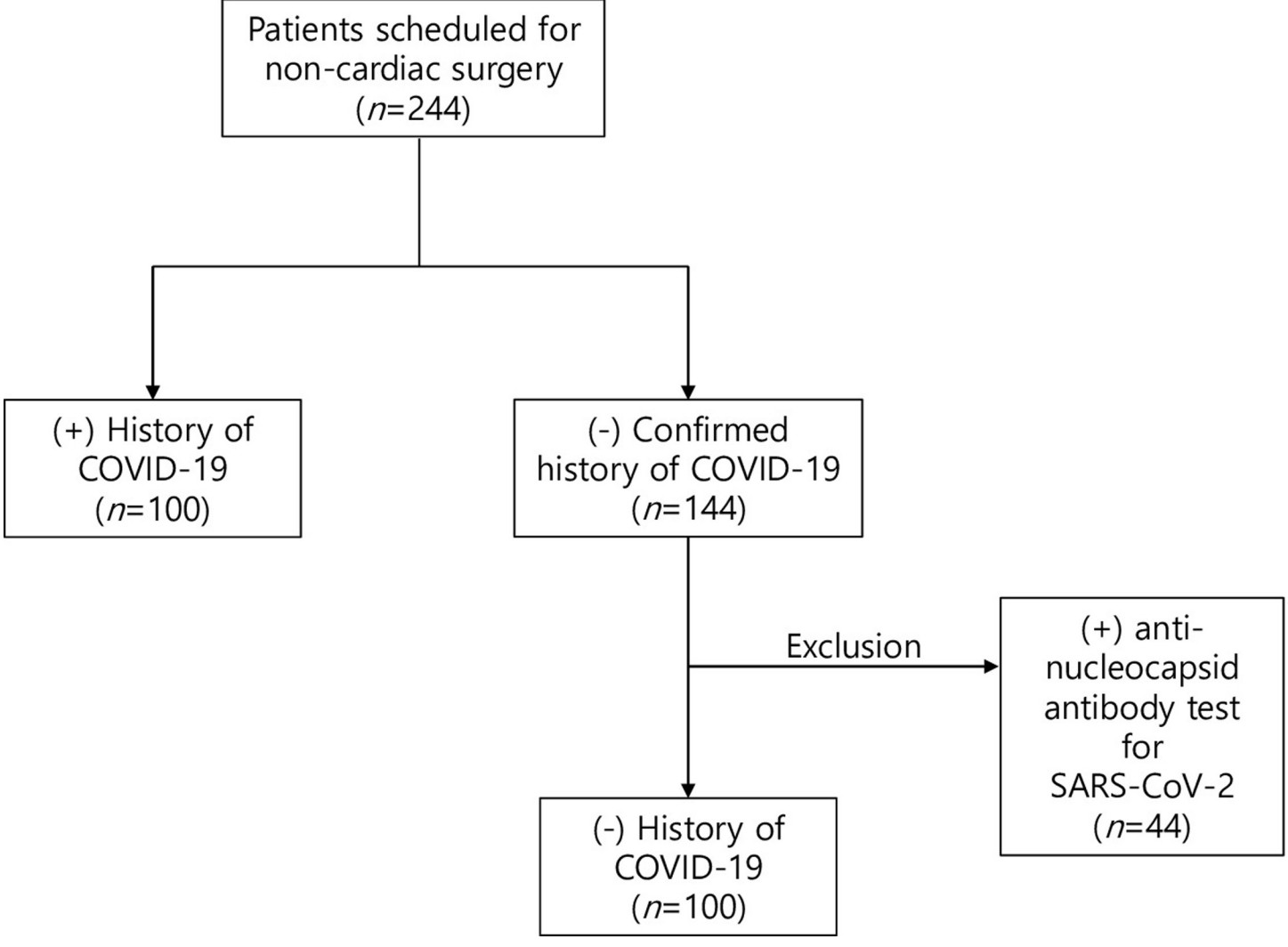

**Fig 1. Patient enrolment flowchart.**

comparative cohort [5, 25]. In addition, undetected infections have rarely been considered in previous studies. The big problem on this topic is that the actual effect of COVID-19 may need to be more precisely evaluated, owing to a gap between prevalence based on self-reported history and a true prevalence using both molecular and serology assessment [28, 29]. This gap may underestimate the true extent of COVID-19 at the population level and misguide future clinical plans. Given these considerations, we excluded the patients with undetected infections using a serologic antibody test and compared the risk of PPCs according to the COVID-19 history. This approach would allow our study to provide more accurate information on the relationship between the history of COVID-19 and the risk of PPCs following major non-cardiac surgery.

We choose PPCs as the outcome of interest in this study because the upper airway is the primary infection site of COVID-19 which can lead to severe respiratory illness requiring intensive care unit hospitalization [30]. Pulmonary symptoms were also focused on in a recent study reporting the lasting symptom burden of COVID-19 patients [31].

Despite previous reports on the devastating impact of SARS-CoV-2 infection on postoperative morbidity and mortality, our patients with a history of no or mild symptomatic COVID-

**Table 1. Baseline characteristics of patients with and without history of coronavirus disease 2019 (COVID-19).**

| | History of COVID-19 (N = 100) | No history of COVID-19 (N = 100) | P-value |
|---|---|---|---|
| Age | 54.84 (±15.04) | 58.20 (±14.06) | 0.1 |
| Male | 46 (46.0) | 43 (43.0) | 0.78 |
| Body mass index | 24.90 (±3.84) | 24.50 (±3.96) | 0.47 |
| ASA physical status | | | 0.20 |
| I | 6 (6.0) | 1 (1.0) | 0.12 |
| II | 79 (79.0) | 84 (84.0) | 0.47 |
| III | 15 (15.0) | 15 (15.0) | >0.99 |
| Smoking status | | | 0.86 |
| Non-smoker | 83 (83.0) | 80 (80.0) | >0.99 |
| Ex-smoker | 8 (8.0) | 10 (10.0) | 0.81 |
| Current smoker | 9 (9.0) | 10 (10.0) | >0.99 |
| Comorbidities | | | |
| Hypertension | 32 (32.0) | 35 (35.0) | 0.77 |
| Diabetes mellitus | 13 (13.0) | 21 (21.0) | 0.19 |
| Stroke | 8 (8.0) | 4 (4.0) | 0.37 |
| Coronary artery disease | 4 (4.0) | 3 (3.0) | >0.99 |
| Chronic kidney disease | 1 (1.0) | 8 (8.0) | 0.04 |
| Heart failure | 0 | 1 (1.0) | >0.99 |
| Chronic obstructive pulmonary disease | 1 (1.0) | 2 (2.0) | >0.99 |
| Tuberculosis | 1 (1.0) | 1 (1.0) | >0.99 |
| Malignancy | 18 (18.0) | 23 (23.0) | 0.48 |
| Operation duration | 226.05 (±142.83) | 195.58 (±123.73) | 0.11 |
| Operation types | | | 0.3 |
| Orthopedic | 10 (10.0) | 5 (5.0) | |
| Head and neck | 7 (7.0) | 5 (5.0) | |
| Abdominopelvic | 21 (21.0) | 15 (15.0) | |
| Vascular | 1 (1.0) | 3 (3.0) | |
| Brain | 61 (61.0) | 72 (72.0) | |

Data are presented as mean (±standard deviation) or number (%). ASA, American Society of Anesthesiologists.

19 showed no increased risk of PPCs. This is probably because majority of our patients underwent surgery at least 8 weeks after SARS-CoV-2 infection. This finding is in line with the previous studies, demonstrating that the surgery within 4 weeks of SARS-CoV-2 infection led to a

**Table 2. Risk of postoperative pulmonary complication according to history of coronavirus 2019 (COVID-19).**

| | History of COVID-19 (N = 100) | No history of COVID-19 (N = 100) | Unadjusted OR (95% CI) | P-value | Adjusted OR (95% CI) | P-value |
|---|---|---|---|---|---|---|
| Postoperative pulmonary complication | 24 (24.0) | 26 (26.0) | 0.95 (0.67–1.31) | 0.74 | 0.99 (0.71–1.37) | 0.92 |
| Atelectasis | 8 (8.0) | 7 (7.0) | 1.08 (0.63–1.82) | 0.79 | 1.10 (0.64–1.89) | 0.73 |
| Pleural effusion | 7 (7.0) | 10 (10.0) | 0.82 (0.50–1.36) | 0.45 | 0.92 (0.54–1.56) | 0.75 |
| Pulmonary edema | 1 (1.0) | 2 (2.0) | 0.70 (0.21–2.36) | 0.57 | 0.88 (0.24–3.24) | 0.85 |
| Pneumonia | 4 (4.0) | 4 (4.0) | 1.00 (0.49–2.03) | >0.99 | 1.08 (0.52–2.25) | 0.85 |
| Aspiration pneumonitis | 0 | 1 (1.0) | | | | |
| Need for oxygen therapy | 18 (18.0) | 15 (15.0) | 1.12 (0.77–1.62) | 0.57 | 1.14 (0.78–1.67) | 0.51 |

Data are presented as n (%). OR, odds ratio.

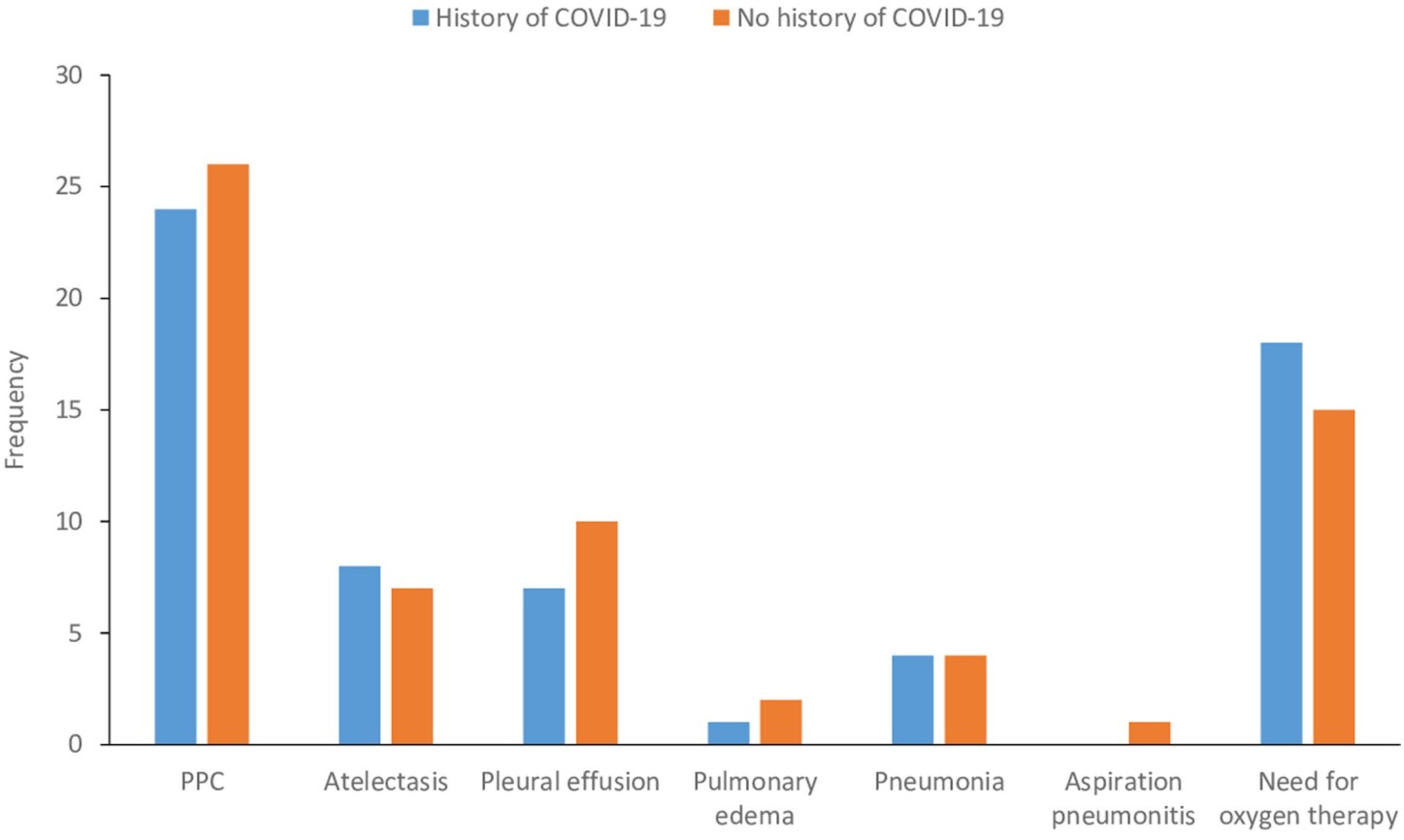

**Fig 2. Incidence of postoperative pulmonary complications according to COVID-19 history.**

higher rate of PPCs, while delaying surgery for approximately 7–8 weeks may reduce the risk of developing PPCs [8, 11]. Therefore, our results support the possibility that surgery after an 8-week period following a mild COVID-19 infection may not directly increase postoperative adverse outcomes. So, paying overdue attention or using additional resources to prevent PPCs in late post-COVID-19 patients with previous mild symptoms may be unnecessary.

Our result should be interpreted considering several limitations as follows: First, we could not assess the incidence of other more severe complications except PPCs because the sample size of this study was estimated using the incidence of PPCs which was relatively higher than that of other complications. Additionally, the actual incidences of PPCs in the current study (24% vs. 26%) were lower than our assumptions (50% vs. 30%) used to calculate the sample size. The lower incidence of PPCs than our expectation may be due to different population or surgical factors compared to our previous study [19]. Considering the lower incidence of PPCs and smaller difference between the groups compared to our assumption, it is possible that our study was underpowered due to overestimation of risk of PPCs. Second, none of our study patients had severe symptoms of COVID-19 and did not undergo surgery within four weeks. Due to the limited nature of our subjects, our results may only be generalizable to some patients with a history of COVID-19. Further research is needed to fully understand the impact of COVID-19 on postoperative outcomes across different severity ranges. Third, the serology assessment was conducted only in patients without a COVID-19 history. Although all COVID-19 was confirmed with RT–PCR test, the antibody was not quantified in these patients. The significance of immunological status of patients with a history of COVID-19

**Table 3. Postoperative pulmonary complication according to relevant factors in patients with a history of coronavirus 2019 (COVID-19).**

| | History of COVID-19 (N = 100) | Incidence of PPC |
|---|---|---|
| Duration from COVID-19 diagnosis to surgery | | |
| < 4 weeks | 4 (4%) | 0 |
| 4 weeks to 8 weeks | 8 (8%) | 12.5% (1/8) |
| > 8 weeks | 88 (88%) | 26% (23/88) |
| Number of vaccinations | | |
| 1 | 2 (2%) | 0 |
| 2 | 28 (28%) | 35.7% (10/28) |
| 3 | 65 (65%) | 21.5% (14/65) |
| 4 | 5 (5%) | 0 |
| Symptoms of COVID-19 | | |
| No symptom | 27 (27%) | 22.2% (6/27) |
| Fatigue | 18 (18%) | 38.9% (7/18) |
| Myalgia | 31 (31%) | 19.4% (6/31) |
| Sore Throat | 42 (42%) | 26.2% (11/42) |
| Cough | 35 (35%) | 25.7% (9/35) |
| Sputum | 23 (23%) | 17.4% (4/23) |
| Fever | 28 (28%) | 25.6% (8/28) |

Data are presented as n (%). PPC, postoperative pulmonary complication.

needs to be considered in future studies. Lastly, perioperative care was standardized but not controlled in a detailed manner.

Despite those limitations, this study accurately classified COVID-19 history through a serology test and identified the actual effect of a COVID-19 history on PPCs in patients who have experienced no or mild symptoms. We emphasize that the perioperative management of patients with a history of COVID-19, especially those with mild or no symptoms, should not be inferred from concerns that a history of SARS-CoV-2 infection increases the risk of PPC. We should acknowledge mild nature of their disease entity and perform a tailored perioperative care for this generalized population undergoing major non-cardiac surgery in the endemic COVID-19 era.

## Conclusion

In our study, patients with a history of no or mild symptomatic SARS-CoV-2 infection did not show an increased risk of PPCs compared to those without a COVID-19 history, which was confirmed by the absence of anti-N antibodies from a serology test. Therefore, additional precautions to prevent PPCs may be unnecessary in those patients.

## Supporting information

**S1 Table. Baseline characteristics and outcomes.**
(DOCX)

**S1 Checklist. TREND statement checklist.**
(PDF)

## Author Contributions

**Conceptualization:** Jong-Hwan Lee.

**Data curation:** Ah Ran Oh, Jungchan Park, Sangmin Maria Lee.

**Formal analysis:** Jungchan Park.

**Methodology:** Eun-Suk Kang, Jungchan Park, Mijeong Jeong.

**Writing – original draft:** Ah Ran Oh, Eun-Suk Kang.

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
