## [Decision Letter · Decision Letter 0]

18 Dec 2023

PONE-D-23-34872Does coronavirus disease 2019 history alone increase the risk of postoperative pulmonary complications after surgery? Prospective observational study using serology assessmentPLOS ONE

Dear Dr. Lee,

Thank you for submitting your manuscript to PLOS ONE. After careful consideration, we feel that it has merit but does not fully meet PLOS ONE’s publication criteria as it currently stands. Therefore, we invite you to submit a revised version of the manuscript that addresses the points raised during the review process.

The paper is well-written, and appropriate statistical methods have been applied to the analyses, with results effectively presented. However, some minor clarifications and revisions are required. I kindly request that you thoroughly review the comments provided by the two reviewers and make the necessary adjustments to the manuscript accordingly.

We look forward to receiving your revised manuscript.

Kind regards,

Yuyan Wang, Ph.D.

Academic Editor

PLOS ONE

Journal Requirements:

Reviewers' comments:

Reviewer's Responses to Questions

**Comments to the Author**

1. Is the manuscript technically sound, and do the data support the conclusions?

Reviewer #1: Yes

Reviewer #2: Partly

2. Has the statistical analysis been performed appropriately and rigorously? 

Reviewer #1: Yes

Reviewer #2: Yes

3. Have the authors made all data underlying the findings in their manuscript fully available?

Reviewer #1: No

Reviewer #2: No

4. Is the manuscript presented in an intelligible fashion and written in standard English?

Reviewer #1: Yes

Reviewer #2: Yes

5. Review Comments to the Author

Reviewer #1: The authors conducted a prospective study to explore whether history of Covid-19 infection could increase the risk of postoperative PPCs in non-cardiac surgeries. The intuition for conducting the study is well-explained, though the focus on mild Covid-infections and ASA I-III surgery patients are not very innovative and inspiring. The statistical analysis and the results are also clearly explained and presented. I only have a few minor comments on the details.

Ln 112. Citation 14 seems unrelated to the specificity of the test.

Ln 150. The sample size was calculated on 30% control group incidence which is higher than the actual incidence. The projected incidence of 50% in the case group seems too pessimistic clinically as well. Sample size is especially sensitive when incidence and effective size are small. Although it will not change the conclusion, the small sample size is a concern, and I would suggest the authors to include post-hoc, a detectable effect size based on the sample size and the actual incidence of PPCs.

Table 1. Overall p-values should be reported for ASA and smoking status.

Reviewer #2: The manuscript is well-structured and articulately composed, effectively encompassing the essential elements of a prospective observational study succinctly. The research questions and hypotheses are presented with commendable clarity. The study's design is logically chosen and aligns well with the objectives of the research. However, I have identified a few areas where minor enhancements could be beneficial:

The criteria for inclusion and exclusion are succinctly stated but could benefit from greater detail. For instance, in lines 104-107, the methods used for determining eligibility could be bolstered by providing supportive citations from existing literature or by elaborating on the rationale for the chosen approach. This additional context would help to clarify the selection process for readers and fellow researchers.

The authors are encouraged to discuss more about the clinical importance of their study. Given the prevailing understanding that mild Covid-19 typically has a minimal impact on patient life, the absence of differences between the study groups might not seem clinically inspiring. Besides, should the authors take the duration of Covid-19 from diagnosis to surgery into consideration? The discussion should address whether the duration from recovery to surgery might serve as a confounding variable that influences the study's outcomes. Specifically, the manuscript should explore the possibility that after an 8-week period following a mild COVID-19 infection, any lingering health sequelae may not be substantial enough to impact surgical risks significantly.

Minor:

Line 103, “also” and “as well” duplicated

The figures are unclear and low pixels

6. PLOS authors have the option to publish the peer review history of their article (what does this mean?). If published, this will include your full peer review and any attached files.

Reviewer #1: No

Reviewer #2: No

---

## [Author Response · Author response to Decision Letter 0]

4 Feb 2024

Reviewer #1

The authors conducted a prospective study to explore whether history of Covid-19 infection could increase the risk of postoperative PPCs in non-cardiac surgeries. The intuition for conducting the study is well-explained, though the focus on mild Covid-infections and ASA I-III surgery patients are not very innovative and inspiring. The statistical analysis and the results are also clearly explained and presented. I only have a few minor comments on the details.

>> Response: Thank you for your kind comments. 

1. Ln 112. Citation 14 seems unrelated to the specificity of the test.

>> Response: We re-checked the reference and changed it to the appropriate reference as follows. Also, the number of citation was changed from 14 to 17.

>> Change: 17. Elecsys® Anti-SARS-CoV-2. Package Insert 2020-07, V3.0; Material Numbers 09203095190 and 09203079190. (Page 17, line 308-309)

2. Ln 150. The sample size was calculated on 30% control group incidence which is higher than the actual incidence. The projected incidence of 50% in the case group seems too pessimistic clinically as well. Sample size is especially sensitive when incidence and effective size are small. Although it will not change the conclusion, the small sample size is a concern, and I would suggest the authors to include post-hoc, a detectable effect size based on the sample size and the actual incidence of PPCs.

>> Response: We greatly appreciate the reviewer for the critical comment. We agree with your opinion that the incidences of PPCs were relatively higher in our retrospective data (30%, COVID (-) group) and our assumption (50%, COVID (+) group). The actual incidences of PPCs in the current study were 24% in COVID-19 (+) group and 26% in COVID-19 (-) group, respectively. This discrepancy probably has resulted from differences in baseline patients’ characteristics and surgical factors between the studies. Unfortunately, we are unable to predict this difference in advance, so our study might seem underpowered (When we performed pos-hoc power analysis based on the actual incidence of PPCs and sample size, the power of this study was less than 0.8). Therefore, we agree with the reviewer’s opinion that we should inform of the possibility that our study was underpowered. We added more details about this problem to the limitations as follows:

>> Change: “First, we could not assess the incidence of other more severe complications except PPCs because the sample size of this study was estimated using the incidence of PPCs which was relatively higher than that of other complications. Additionally, the actual incidences of PPCs in the current study (24% vs. 26%) were lower than our assumptions (50% vs. 30%) used to calculate the sample size. The lower incidence of PPCs than our expectation may be due to different population or surgical factors compared to our previous study. Considering the lower incidence of PPCs and smaller difference between the groups compared to our assumption, it is possible that our study was underpowered due to overestimation of risk of PPCs.” (Page 14, line 234-242)

3. Table 1. Overall p-values should be reported for ASA and smoking status.

>> Response: Following your recommendation, we reported overall P-values for ASA physical status and smoking status in Table 1. There were no significant differences between the two groups. Please check the revised version of Table 1. 

Reviewer #2

The manuscript is well-structured and articulately composed, effectively encompassing the essential elements of a prospective observational study succinctly. The research questions and hypotheses are presented with commendable clarity. The study's design is logically chosen and aligns well with the objectives of the research. However, I have identified a few areas where minor enhancements could be beneficial:

>> Response: Thank you for your kind comments.

1. The criteria for inclusion and exclusion are succinctly stated but could benefit from greater detail. For instance, in lines 104-107, the methods used for determining eligibility could be bolstered by providing supportive citations from existing literature or by elaborating on the rationale for the chosen approach. This additional context would help to clarify the selection process for readers and fellow researchers.

>> Response: Following your recommendation, we cited previous studies that used similar methodology to this study to identify COVID-19 history accurately. Also, we have added more details about IgG antibodies as follows to further clarify the patient selection protocol.

>> Change: “We also identified patients without history of COVID-19 as having negative result in immunoglobulin G (IgG) test for SARS-CoV-2 nucleocapsid antigen (anti-N antibody) [14, 15]. The detection of SARS-CoV-2 IgG antibody is an indicator of recovery and immunity from COVID-19, which implies that the patient was previously infected with SARS-CoV-2 [16]. For all patients reporting no history of confirmed diagnosis of COVID-19, we tested anti-N antibody using 3ml sample of blood during intraoperative period to identify undetected infection in the past. Then, only seronegative patients were included and classified as cases without a history of COVID-19.” (Page 6 , line 103 – Page 7, line 110)

14. Jungwirth-Weinberger, A., et al., D-dimer levels are not elevated in SARS-CoV-2 IgG positive patients undergoing elective orthopedic surgery. Journal of Clinical Medicine, 2021. 10(16): p. 3508.

15. Jungwirth-Weinberger, A., et al., History of COVID-19 was not associated with length of stay or in-hospital complications after elective lower extremity joint replacement. Arthroplasty today, 2022. 13: p. 109-115.

16. Hou, H., et al., Detection of IgM and IgG antibodies in patients with coronavirus disease 2019. Clinical & translational immunology, 2020. 9(5): p. e1136.

2. The authors are encouraged to discuss more about the clinical importance of their study. Given the prevailing understanding that mild Covid-19 typically has a minimal impact on patient life, the absence of differences between the study groups might not seem clinically inspiring. Besides, should the authors take the duration of Covid-19 from diagnosis to surgery into consideration? The discussion should address whether the duration from recovery to surgery might serve as a confounding variable that influences the study's outcomes. Specifically, the manuscript should explore the possibility that after an 8-week period following a mild COVID-19 infection, any lingering health sequelae may not be substantial enough to impact surgical risks significantly.

>> Response 1: We fully agree with your opinion that we need to clarify the clinical implication of our research, especially why it is important to focus on the mild COVID-19 patients. Although it has been reported a significant increase in morbidity and mortality for perioperative SARS-CoV-2 positive patients, there is little evidence of whether these patients still have an increased risk of surgical complications after complete recovery from mild SARS-CoV-2 infection. In our results, we found that patients recovering from mild COVID-19 were not associated with increased PPCs after non-cardiac surgery. Therefore, our study emphasizes that their perioperative management should not be inferred from other COVID-19 patients but should be addressed independently. For further clarify, we added clinical importance of our study as follows in discussion section.

>> Change 1: “Despite those limitations, this study accurately classified COVID-19 history through a serology test and identified the actual effect of a COVID-19 history on PPCs in patients who have experienced no or mild symptoms. We emphasize that the perioperative management of patients with a history of COVID-19, especially those with mild or no symptoms, should not be inferred from concerns that a history of SARS-CoV-2 infection increases the risk of PPC. We should acknowledge mild nature of their disease entity and perform a tailored perioperative care for this generalized population undergoing major non-cardiac surgery in the endemic COVID-19 era.” (Page 15, line 252-259)

>> Response 2: Also, we agree with you that we should clarify why we considered time of surgery as a confounding factor for this study. According to previous studies, it has been reported that the risk of PPCs increases significantly when the surgery was conducted within 4 weeks of SARS-CoV-2 infection but decreased after 7-8 weeks of infection (Please refer the references 8,11). Therefore, we considered it necessary to evaluate whether time of surgery may also affect the risk PPCs in our patients. We clarified this and added its clinical importance in the Discussion section as follows. 

>> Change 2: “Despite previous reports on the devastating impact of SARS-CoV-2 infection on postoperative morbidity and mortality, our patients with a history of no or mild symptomatic COVID-19 showed no increased risk of PPCs. This is probably because majority of our patients underwent surgery at least 8 weeks after SARS-CoV-2 infection. This finding is in line with the previous studies, demonstrating that the surgery within 4 weeks of SARS-CoV-2 infection led to a higher rate of PPCs, while delaying surgery for approximately 7-8 weeks may reduce the risk of developing PPCs [8, 11]. Therefore, our results support the possibility that surgery after an 8-week period following a mild COVID-19 infection may not directly increase postoperative adverse outcomes. So, paying overdue attention or using additional resources to prevent PPCs in late post-COVID-19 patients with previous mild symptoms may be unnecessary.” (Page 14, line 223-233)

Minor:

3. Line 103, “also” and “as well” duplicated

>> Response: Following your recommendation, we revised the sentence as follows.

>> Change: “We also identified patients without history of COVID-19 as having negative result in immunoglobulin G (IgG) test for SARS-CoV-2 nucleocapsid antigen (anti-N antibody).” (Page 6, line 103-105)

4. The figures are unclear and low pixels

>> Response: We updated the figures to a clear version. Please check Figure 1 and 2.

---

## [Decision Letter · Decision Letter 1]

6 Mar 2024

Does coronavirus disease 2019 history alone increase the risk of postoperative pulmonary complications after surgery? Prospective observational study using serology assessment

PONE-D-23-34872R1

Dear Dr. Lee,

We’re pleased to inform you that your manuscript has been judged scientifically suitable for publication and will be formally accepted for publication once it meets all outstanding technical requirements.

Kind regards,

Yuyan Wang, Ph.D.

Academic Editor

PLOS ONE

Additional Editor Comments (optional):

Reviewers' comments:

Reviewer's Responses to Questions

**Comments to the Author**

1. If the authors have adequately addressed your comments raised in a previous round of review and you feel that this manuscript is now acceptable for publication, you may indicate that here to bypass the “Comments to the Author” section, enter your conflict of interest statement in the “Confidential to Editor” section, and submit your "Accept" recommendation.

Reviewer #1: All comments have been addressed

Reviewer #2: All comments have been addressed

2. Is the manuscript technically sound, and do the data support the conclusions?

Reviewer #1: Yes

Reviewer #2: Yes

3. Has the statistical analysis been performed appropriately and rigorously? 

Reviewer #1: Yes

Reviewer #2: Yes

4. Have the authors made all data underlying the findings in their manuscript fully available?

Reviewer #1: Yes

Reviewer #2: Yes

5. Is the manuscript presented in an intelligible fashion and written in standard English?

Reviewer #1: Yes

Reviewer #2: Yes

6. Review Comments to the Author

Reviewer #1: (No Response)

Reviewer #2: (No Response)

7. PLOS authors have the option to publish the peer review history of their article (what does this mean?). If published, this will include your full peer review and any attached files.

Reviewer #1: No

Reviewer #2: No

---

## [Editor Report · Acceptance letter]

2 May 2024

PONE-D-23-34872R1 

PLOS ONE

Dear Dr. Lee, 

I'm pleased to inform you that your manuscript has been deemed suitable for publication in PLOS ONE. Congratulations! Your manuscript is now being handed over to our production team.

Kind regards, 

on behalf of

Dr. Yuyan Wang 

Academic Editor

PLOS ONE